# Effects of Oro-Sensory Exposure on Satiation and Underlying Neurophysiological Mechanisms—What Do We Know So Far?

**DOI:** 10.3390/nu13051391

**Published:** 2021-04-21

**Authors:** Marlou P. Lasschuijt, Kees de Graaf, Monica Mars

**Affiliations:** Division of Human Nutrition and Health, Wageningen University & Research, P.O. Box 17, NL-6700 AA Wageningen, The Netherlands; marlou.lasschuijt@wur.nl (M.P.L.); kees.degraaf@wur.nl (K.d.G.)

**Keywords:** satiation, food intake, taste, texture, oro-sensory exposure, sensory science, cephalic responses, brain areas, brain stem, weight management

## Abstract

The mouth is the first part of the gastrointestinal tract. During mastication sensory signals from the mouth, so-called oro-sensory exposure, elicit physiological signals that affect satiation and food intake. It has been established that a longer duration of oro-sensory exposure leads to earlier satiation. In addition, foods with more intense sweet or salty taste induce earlier satiation compared to foods that are equally palatable, but with lower taste intensity. Oro-sensory exposure to food affects satiation by direct signaling via the brainstem to higher cortical regions involved in taste and reward, including the nucleus accumbens and the insula. There is little evidence that oro-sensory exposure affects satiation indirectly through either hormone responses or gastric signals. Critical brain areas for satiation, such as the brainstem, should be studied more intensively to better understand the neurophysiological mechanisms underlying the process of satiation. Furthermore, it is essential to increase the understanding of how of highly automated eating behaviors, such as oral processing and eating rate, are formed during early childhood. A better understanding of the aforementioned mechanisms provides fundamental insight in relation to strategies to prevent overconsumption and the development of obesity in future generations.

## 1. Introduction

Eating is an episodic process, people consume meals or snacks followed by periods of fasting. People eat of a meal until they feel satiated, which translates into satiety, i.e., the period between two eating episodes. The processes of satiation and hunger involve feedback mechanisms that originate from different parts of the gastro-intestinal tract. Ingested and absorbed food components elicit a cascade of physiological and metabolic processes that affect satiety and, consequently, food intake [1,2]. Oral signals and perceived sensory properties of foods have been considered to play a role in the early, pre-absorptive phase of food ingestion. Post-absorptive factors, arising from the stomach and intestines, play a role in satiety and the initiation of the next meal. In this review, we focus on the physiological processes that are initiated as a result of the presence of food in the mouth, i.e., oro-sensory signals. This review addresses how oro-sensory signals relate to earlier satiation and lowering food intake, in the context of weight management and obesity prevention.

In textbooks, food digestion is often referred to as the digestive process that starts in the stomach and continues in the intestines. However, food digestion already starts in the mouth and it has an essential function in food digestion. The mouth has a gate-keeping function as it prevents ingestion of unfamiliar, bad-tasting and potentially dangerous food. Additionally, it prepares the body for food digestion further along the digestive tract. In the mouth, food is masticated to be mechanically broken down into small pieces, which enlarges the surface area. The food is mixed and lubricated with α-amylase containing saliva to form a bolus. This allows for the start of carbohydrate digestion in the mouth and makes the bolus safe to swallow [3]. During this mastication process, food is perceived in the mouth by the senses (mostly taste and retro-nasal smell) and multiple physiological processes are initiated (for reviews see [4,5]). Eating food is potentially a threat to the body’s homeostasis, therefore, oral exposure serves as an alarm such that the body can anticipate and prepare incoming nutrients for optimal digestion [6,7]. 

The role of oral exposure to food (taste, flavor, texture) has been extensively studied in relation to satiety and food intake. One of the most prominent findings in this area is a phenomenon now referred to as “sensory-specific satiety”. Sensory-specific satiety was first described by Rolls et al. in 1981, where it was defined as the decrease in pleasantness of the eaten food relative to the pleasantness of foods not eaten [8]. It was found that if people were instructed to eat a certain food until satiation, and then were provided with a series of different tasting foods, the rating of pleasantness of the just eaten foods was decreased more compared to the non-eaten foods. This decrease in pleasantness could be explained primarily by the sensory characteristics of the food, rather than sensory fatigue or the nutrient composition of the food. This observation establishes that sensory exposure to food directly affects its hedonic value while being consumed to satiation.

In this review, we reflect on the scientific evidence that demonstrates the relation between oro-sensory exposure and the regulation of food intake, with a focus on satiation (within meal satiety). In addition to the different components of oro-sensory exposure and their effects on food intake, we discuss hypothesized underlying neuro-physiological mechanisms (Figure 1). We describe main knowledge gaps and directions for future research. Finally, the implications for obesity prevention are discussed.

## 2. Effects on Satiation

### 2.1. Factors Affecting Oro-Sensory Exposure

Oro-sensory exposure reflects the overall exposure of the oral cavity to sensory cues during food consumption [10,11]. The level of oro-sensory exposure, also called “in-mouth sensory exposure”, is determined by both the sensory properties of a food and its processing in the mouth (oral processing). The magnitude of sensory exposure is determined by (1) the intensity of the stimulus and (2) the duration of exposure to the stimulus (see Figure 2). The duration and intensity of oro-sensory exposure are not two distinct factors, but often interact, especially in food with complex textures where oral processing affects the release of tastants and vice versa. Furthermore, the nature of the stimulus, i.e., the sensory quality or modality of the ingested food, is important. In this review, we focus on texture and taste as being the main qualities of importance for oro-sensory exposure. Taste and texture affect the duration, quality, and intensity of oro-sensory exposure which affects satiation and, consequently, the amount of food consumed. In the following paragraphs, we reflect on the evidence for an effect of oro-sensory exposure on food intake, starting with the duration of oral exposure.

### 2.2. Oral Exposure Duration

The duration of oral exposure is determined mainly by the texture of food and the rate at which it is eaten. Different textures elicit different oral processing behavior and may therefore affect oro-sensory exposure time to the food. There is a wide variation in eating rate of foods available in our diet [12,13], and in how people process food in mouth [14]. Accordingly, oral processing time is in part determined by individual eating behavior, but for a larger part, by the food’s textural properties [13,15].

Texture can be defined as the multimodal sensory experience derived from the structure of food. It can be characterized by terms such as viscous, chewy, elastic, and hard [16]. Food texture may also be classified into broader groups of food form: liquid, semi-solid, soft solid, and hard solid [17]. Furthermore, there are mixed structures, for example hard particles added to semi-solids, can change oral processing behavior [18,19].

In the last 15 years, many experimental studies have shown consistently that food form affects how much people eat, independent of the palatability of the food. Liquids have a lower satiation capacity relative to their energy content compared to semi-solids and solids. This is due to the high eating or drinking rate at which liquids can be consumed which induces short oro-sensory exposure [20]. For example, Zijlstra et al. reported that food intake decreased when the viscosity of chocolate milk was increased to a custard like product [15]. This observation has been replicated several times [21,22]. It has also shown that this effect is mainly due to oral exposure time [15,23]. Eating the same semi-liquid with a food with a straw (short oral exposure) compared to eating with a spoon (longer oral exposure) resulted in a significantly lower intake [23]. Accordingly, eating foods in a way that it yields longer oral exposure leads to earlier satiation and consequently lower food intake.

Within the group of semi-solid and solid foods, there appears to be a window of opportunity to change satiating capacity by changing the food structure and thus oral processing. For example, mashed potatoes are less satiating compared to potatoes with their full structure [24] and hamburgers with crispy hard bread compared to soft bread [25]. However, the changes in oral processing that are elicited by texture changes should be large enough to alter the rate at which the foods are eaten. For example, Zijlstra et al., compared different textures of three solids foods, candies, meat, and meat replacers, and found the changes that were elicited in oral processing were not large enough to result in effects in food intake [26]. Recently Chayadi et al., failed to show an effect between two types of crisps with different textures [27]. In a recent review, Bolhuis and Forde provided an elaborate overview with food structure manipulations that can potentially lead to foods that have a high satiating capacity [28].

In addition to changing oral exposure time indirectly by altering the texture of foods, it has also been shown that manipulating oro-sensory exposure duration directly affects intake. For example, by manipulating sip sizes by using peristaltic pumps, or instructions during eating. For example, Weijzen et al. manipulated sip sizes of sweet orangeades with a peristaltic pump; drinking small sips (5 g) compared to large sips (20 g) resulted in a lower intake [29]. However, this effect was only statistically significant for sugar-sweetened drinks and not when the orangeades were sweetened with non-caloric sweeteners. This indicates that a metabolic reward is involved in this effect. Similarly, Zijlstra et al. did an experiment in which they manipulated bite-size and oral processing time of chocolate milk with a peristaltic pump [21]. Instructing individuals to eat slower and process foods longer in their mouths resulted in smaller intakes, also compared to the “free eating conditions” in which individuals could choose their own bite size and oral processing time. Also, more real-life experiments have also shown this effect. For example, Weijzen et al. instructed individuals to either eat nibble-sized or bar size snacks, and either eat them attentively or without any specific instruction [30]. Subjects ate less of the nibbles compared to the bars. Interestingly, they only found a clear effect of nibbles and bars on food intake but no convincing effect of eating with more attention.

In summary, it has been shown in numerous studies that longer oro-sensory exposure by processing foods longer in the mouth, and thus eating slower, decreases meal size. This effect appears to be independent of palatability or the degree of attention to food.

### 2.3. Taste Intensity and Quality

The intensity of oro-sensory exposure intensity is reflected by the perceived intensity. The latter is determined by the binding of the tastant to the taste receptors in the oral cavity and tongue. This is dependent on the concentration of the tastant, as well as on the surface availability and its solubility in saliva [31]. Surface availability and the capacity of tastants to dissolve in saliva are both dependent on food texture. For example, two foods with the same amount of sucrose, but differing in texture, (liquid vs. solid), have different sweetness intensities [32]. For a solid food product to be equally sweet to a liquid, twice the amount of sucrose is needed [33,34].

The taste intensity of food affects satiation and subsequent food intake. Accordingly, smaller meals are consumed when foods of higher sweetness when provided *ad libitum* [35,36,37] even if the rated palatability of the two products before ingestion was the same. This is also the case for salt intensity. Bolhuis conducted a series of experiments in which salt intensity was varied; tomato soups with similar palatability, but slightly higher and lower than ideal salt intensities were provided and intake was measured [11,38]. Subjects consumed less soup in the high compared to the low salt intensity. Therefore it appears that exposure to higher sweetness intensity and higher saltiness intensity leads to earlier meal termination and a reduction in food intake. 

It may be argued that this effect is only specific for sweetness and saltiness, and that the magnitude may differ between tastes. However, to our knowledge, there is only one study that investigated whether exposure to different taste qualities has different effects on satiation. Griffioen, et al. offered two equally liked rice meals, with similar structure and macronutrient content, one with sweet taste compared to savory rice meals, and found no difference in *ad libitum* consumption of these two meals, that is they were equally satiating [39].

The concentration of a tastant in a food product does not predict intensity per se as the release of tastes and flavors is also dependent on the structure of the food. It has been shown that a harder structure enhances the effect of intensity, by prolonging the exposure duration (i.e., harder textures require longer oral processing before a bolus is safe to swallow) [24,40]. For example, Forde et al. reported that a higher savory taste, in combination with a harder texture of the food, led to a significant decrease in food intake compared to non-flavored soft food [24]. While Lasschuijt et al. reported that a combination of a harder texture and higher sweetness intensity leads to a reduction in intake. The latter study also showed that the effect of texture was much larger compared to the effect of sweetness intensity [40].

### 2.4. Discussion Effects of Oro-Sensory Exposure on Satiation

To summarize, it has been established that oro-sensory exposure has an important role in satiation. The taste intensity of foods and how they are processed in the mouth affect the amount of food people consume, independently of the initial palatability of the food. A review by McCrickerd et al. argues that the effect of texture on food intake is approximately 30% and that of taste 10%, depending on the differences in texture hardness and taste intensity between two foods [41].

In this review, we have only focused on within meal satiation. Although we did not discuss the evidence here, we also know that food form (liquid vs. solids) affects appetite and food intake beyond a single meal [42]. Similarly, taste and flavor of a particular meal may also affect appetite and satiety after the meal [43]. For example, after eating a savory meal, there is a preference for sweet food items for the following meal or snack [43]. This effect on satiety may reflect both direct and indirect signaling related to the entry of food entering the stomach and intestines.

The effect of texture on satiation appears to be universal effect, so independent of eating habits, that is familiarity with the food. We have shown in the past that the effect of texture on satiation exists both for familiar as well as unfamiliar foods [44,45]. Moreover, it has also been shown in Western and Asian populations that texture modifies oral processing consistently across different populations or cultures [42,46,47]. Although between population groups (sex, ages, ethnicity) the effect sizes may differ, the direction of the effects is identical within an individual (lower eating rate and lower intake with harder textures) [48].

Equal palatability of the foods is essential in studies investigating oro-sensory exposure and food intake. It is possible to investigate. You can study two suboptimal perceived intensities of sweet and salt taste that are equally liked [11,36]. However, for the other sensory qualities this is much more difficult to control. For example, we do not know whether bitterness, sourness, or sensory sensations such as trigeminal taste (for example pepper, carbonation, mint) have similar effects on food intake, as the capacity to manipulate these sensory sensations is limited, without foods becoming unpalatable. A number of studies have been done on retro-nasal odors, that is the odor released during mastication, and satiation, but have shown little effect on food intake [46,47].

One of the proposed hypotheses to explain these potent effects of oro-sensory exposure on food intake is that taste and texture are strong predictors of the nutrient content of the food, such as sugar and salt [49]. For example, solid foods have low water content and are therefore more energy-dense. Furthermore, low taste intensity signals low nutrient density. Because of this, solid high taste intensity foods may be perceived as being more rewarding and satiating [50]. An alternative mechanism by which the intensity of oro-sensory exposure may affect satiation is though sensory-specific satiety; strong taste intensities may induce this on more rapidly compared to low taste intensities [10]. This decrease in hedonic value of a taste may lead to meal termination. It has been hypothesized that this phenomenon of sensory specific satiety is essential for survival as it stimulates variety in the diet [49].

## 3. Neurophysiological Mechanisms

The precise physiological mechanisms by which oro-sensory exposure affects satiation has not been elucidated. It has been hypothesized that sensory signals travel from the oral cavity through afferent cranial nerves V VII, IX, X, and XII to the brainstem, and then higher cortical regions, as discussed in Section 3.2. In addition, oro-sensory exposure signals induce a cascade of efferent responses responsible for optimal food digestion, as discussed in Section 3.3 [51]. Ultimately, satiation evolves in the brain (among other regions in the reward center and orbitofrontal cortex) as a result of the integration of direct oro-sensory signals and indirect signals arising from the gastrointestinal tract. The following paragraphs discusses these hypothesized mechanisms in more detail.

### 3.1. Brain Stem

There have been many neuroimaging studies comparing brain responses in hungry and satiated state, however little is known about the brainstem in relation to sensory cues and satiation. The brainstem is the first part of the brain that processes afferent taste and gastric signals, but only limited research has been done in this area [52,53,54]. The brainstem is inherently difficult to investigate because it is a complex area and the cluster size of nuclei with the same function is very small (*n* = 2–5). Even within these small nuclei clusters, nuclei may have very distinct roles. For example, a study in monkeys reported that the posterior part of nuclei of the solitary tract (NTS) is most responsive to glucose, whereas the anterior NTS was particularly responsive to NaCl [55]. This shows the heterogeneity of nuclei even within these small brainstem areas. 

The brainstem is inherently challenging to study. The current brain stem atlases show lack of detail and do not cover all different nuclei in the brainstem. In addition, the brainstem area is difficult to probe as the current fMRI approaches have only limited resolution. As the brainstem is close to the heart and lungs, fMRI signals are prone to artefacts and confounding by physiological noise.

Due to these technical challenges few human studies have been performed; the majority of evidence is derived from animal studies. Based on this work, the dorsal vagal complex (DVC) in the medulla and the parabrachial nucleus (PBN) in the pons play key roles in satiation. The DVC includes the nucleus of the solitary tract (NTS), the dorsal motor vagal nucleus (DMV), and area postrema (AP). Together with the PBN the DVC is involved in the integration of energy-related peripheral signals and processing of taste and gastric signals [56,57,58,59,60]. These brainstem areas receive satiation-related input at three levels. First, neurons sense circulating metabolites and hormones such as glucose and CCK (DMV). Second, the brainstem receives vagal input from the sensory system and the gastrointestinal tract, such as taste (NTS, PBN) and gastric distention signals (PBN). Finally, the brainstem receives input from the mid- and forebrain, which is integrated together with energy-related signals in the area postrema (AP). The brainstem then projects these signals to hepatic and gastric efferent nerves [61]. An overview of the brainstem areas involved in satiation is shown in Figure 3.

As discussed before only a couple of human studies have investigated taste processing and satiation in the brainstem. For example, Small et al. reported that greater taste intensity led to increased activation in the PBN region of the pons [63]. Other studies have found NTS activation after oro-sensory exposure to sour-sweet-salt-bitter mixtures [64] and sucrose solutions [65]. The most convincing evidence for brainstem involvement in the process of satiation comes from a study in rats. Animals that were decerebrated rejected taste stimuli when satiated [66], indicating that despite the absence of functional higher cortical regions, taste signals are still processed in relation to satiation in the brainstem (PBN). 

### 3.2. Higher Cortical Regions

From the brainstem oro-sensory signals diverge into an affective and sensory pathways [61]. The affective pathway includes projections to the hypothalamus, amygdala, parahippocampal gyrus, orbitofrontal cortex (OFC), striatum, and midbrain. The sensory pathway includes the thalamus and insula. Together, these areas regulate the release of hormones involved in digestion and satiety [60,67].

The most prominent region involved in satiation, is the reward system (areas that are part of the affective pathway). With increasing fullness, the reward system is downregulated which lowers the hedonic responses to food (due to direct and indirect signals induced by food intake) leading to meal termination [68].

Other relevant brain regions are the insula and the amygdala. It has been shown that internal state modulates the activation of the insula and that this is associated with a reduction in affective value or pleasantness of the taste stimuli, specific satiation [63,67,69]. The amygdala is associated with the perception of both positive and negative verbal, visual, odor and taste stimuli [70,71,72,73,74,75]. Therefore, the amygdala may signal both pleasure and aversion of taste stimuli, dependent on the satiated state. This is in line with the observations of La Bar et al. and Morris et al. of enhanced responses to food pictures in a hungry state compared to the satiated state [76,77]. However, observations have shown to be inconsistent as some studies reported that activation of the amygdala reflects odor and taste stimulus intensity, irrespective of liking [63,78].

### 3.3. Cephalic Phase Responses

In addition to a direct effect on higher cortical regions, oro-sensory exposure also affects food intake indirectly through so-called cephalic phase reflexes, or cephalic phase responses (CPRs) [5,6,8]. As mentioned, these are induced when oro-sensory signals reach the brainstem, and thought to support optimal food digestion.

CPRs are anticipatory and conditioned responses to food cues [5,6]. They occur within minutes after food consumption and are, therefore, not related to nutrient sensing further down in the digestive tract [4,5]. Cephalic responses include increased production of saliva, gastric juice, secretion of bile by the gallbladder, increased gut motility, and gastric and pancreatic hormone secretion [79,80,81,82]. Examples of cephalic hormones are insulin, pancreatic polypeptide, and ghrelin [4,5,83]. The supposed function of CPRs is to “prepare” the body for incoming nutrients to optimize digestion [6,84]. 

Until recently, a reduction in CPRs was thought to be associated with impaired appetite regulation and weight gain [85,86,87]. Based on this, we hypothesized that a reduction in CPRs from little oro-sensory exposure may represent physiological mechanism underlying reduced satiation—for example, when foods with little taste are eaten quickly [88,89]. Recently, we performed a systematic and quantitative review which investigated the magnitude of the cephalic response of insulin and pancreatic peptide. We concluded that the cephalic insulin and pancreatic responses are almost indistinguishable from normal variation, and exhibit substantial variation in both magnitude and timing of onset. Additionally, only two studies found an effect of a reduced CPRs on subjective appetite ratings [83,90]. Accordingly, it should be questioned whether these CPRs affect satiation and are biologically meaningful in daily life [4]. If cephalic phase responses play an important role in satiation, it is likely to be caused by cephalic phase responses other than insulin and pancreatic polypeptide response, such as early production of saliva and gastric juice [69].

### 3.4. Discussion Neurophysiological Mechanisms in Satiation

To summarize, the neuro-physiological mechanisms underlying satiation are gradually becoming clearer as research progresses. In particularly, neuroimaging studies have provided insight into the processing of sensory signals in the higher cortical regions of the brain in relation to satiation. However, the precise neurophysiological mechanisms of how oro-sensory signals affect the process of satiation, and the role of the brain stem remains to be elucidated.

As mentioned before, neuroimaging of the brain stem is inherently difficult. To improve research in this area, a brainstem atlas suitable for fMRI and optimized normalization of the brainstem is needed. A brainstem atlas would allow for distinction between sensory responses, physiological processes and movement. For example, discriminating tongue movements from taste perception activation. The development of a brainstem atlas would also allow future studies to measure brainstem areas specifically involved in sensory processing, gastric distention, and satiation.

In addition to neuro-physiology, research in the area of the peripheral physiological mechanisms underlying satiation could progress by having a clear definition of cephalic phase responses. Such a definition should take baseline variation of the cephalic variable into consideration to conclude whether a peak increase reflects the presented food cue or biological fluctuations. Furthermore, the definition should include a strict time range in which the cephalic peak response should occur, for example within 2–5 min after food ingestion. Such a clear definition would be helpful in combining and comparing results of different studies which investigate early responses to food cues in relation to satiation.

In this review we have focused solely on the effects of oro-sensory exposure on satiation. However, as shown in Figure 1, both the oral as well as the gastric phase are important. For example, factors in the gastric phase that contribute to satiation are meal volume, meal weight, and, indirectly, gastro-intestinal hormones that inhibit gastric emptying [91,92]. Also gastric emptying is affected by food texture; it is more difficult for digestive fluids to penetrate hard textured foods and are therefore their biochemical breakdown is much slower [93]. Moreover, larger solid particles require more mechanical breakdown, that is muscle movements of the stomach, to reduce the particle size to a level appropriate for entering the duodenum [94]. When gastric emptying rate is low, stomach distention activates stretch and mechanoreceptors send fullness signals to the brain via vagal afferent nerves to the brainstem [95]. Simultaneously, nutrients that enter the small intestine trigger GI hormones including CCK, PYY, and GLP-1 that reach the brain through the blood brain barrier [96,97]. These postprandial hormonal responses are substantially greater than the before mentioned CPRs. It is believed that postprandial responses to account for 40–70% of the variation in predicted food intake [98,99]. However, as these responses occur are mainly elicited after consumption, they are more likely to affect satiety (i.e., between meal suppression of appetite) than the amount eaten within a meal, i.e., satiation. 

## 4. Recommendations for Future Research

Eating slowly due to food texture, or because of individual eating behavior decreases food intake compared to more rapid food intake [20,100]. Oral processing such as chewing behaviour and eating rate differs not only between food structures, but also between individuals. Eating behavior is dependent on gender, age, ethnicity and body weight status [20,100]. Why some people simply eat faster than others is not fully understood. Although eating rate differs between people, within-subject eating speed is a relatively steady and automated behavior. The same person eating the same food product twice results in a similar eating rate. This constant or fixed eating rate appears to be acquired at a young age. Eating rate is in part inheritable, and acquired through learning as children’s eating rate is affected by parental feeding strategies and early exposure to foods with hard texture [101,102]. There is relatively little information on the etiology of children’s eating behavior but it is important to better understand how eating behaviour is formed, a more rapid eating rate at a young age is predictive of adult BMI [103,104,105]. 

Apart from the etiology of eating behaviour there is little known about the physiology of satiation, i.e., the process of going from a hungry to full state. In particular, information about the brainstem is needed to understand how oro-sensory exposure affects satiation. The brainstem is of particular interest as this is where the sensory signals first arrive in the brain to signal incoming food. To research this area a specific human brainstem atlas needs to be developed. 

With respect to the effects of oro-sensory exposure on satiation, the largest body of evidence relates to the effects of texture or eating duration and on the effect of salty and sweet taste. Few studies have investigated other taste effects on satiation, such as bitter, sour, and umami. Taste intensity is difficult to manipulate while maintaining palatability, and energy density-, and macronutrient composition equal. Future research studies should improve the definition of the manipulation of interest while minimizing potential confounding due to changes in other sensory properties or energy density. Chemical and texture analyses, as well as sensory panel evaluations of the study products should be done routinely to better describe the manipulated products.

Besides much controlled ‘sensory’ studies in healthy individuals, research is required in overweight or obese individuals to determine whether a decrease in food intake leads due to texture manipulations results in actual weight loss. It is possible that overweight or obese individuals may be less sensitive to internal food cues and when eating highly palatable foods the effect of oro-sensory exposure duration on satiation may be overruled by hedonic eating [106]. Another strategy may be to determine whether recommending eating foods with hard texture as part of a low-calorie diet may support weight loss in individuals who are overweight or obese. Such an intervention may potentially be considered successful at 10% weight loss, although 5% sustained weight loss already reduces the risk of cardiovascular diseases and type II diabetes [107,108,109,110]. 

## 5. Conclusions

Oro-sensory exposure to food has a direct effect on satiation by direct signaling via the brainstem to higher cortical regions involved in taste and reward, including the nucleus accumbens and the insula. So far there is little evidence that oro-sensory exposure affects satiation indirectly through hormone responses and gastric signals (Figure 4). Critical brain areas involved in satiation, such as the brainstem, must be studied more intensively to better understand the neurophysiological mechanisms underlying satiation. Furthermore, it is crucial to better understand how highly automated eating behaviors, such as oral processing and eating rate, develop during early childhood. Insights into the aforementioned mechanisms will provide novel strategies to prevent overconsumption and the further increases in the bodyweights of future generations.

## 6. Implications

Based on the research referred to in this review it appears clear that oro-sensory exposure should play an important role in strategies to maintain healthy body weight. Harder food structures and a slow eating rate can be used as a strategy to prevent overeating due to the high energy density and palatability of the foods in our food environment. It appears that not all calories are the same, foods with the same caloric content can have markedly different satiating capacities. To prevent overeating and consequent weight gain, it may be beneficial to foods that have a high satiation/energy density ratio. For example, calculating the satiation/energy ratio for a standard portion of popcorn (25 g, 75 kcal), with an average eating rate of 5 g per minute the ratio is 75/5 = 15 kcal/min. This is a rather slow ingestion rate of energy due to the high air content of popcorn. Almonds have a much harder texture compared to popcorn but a higher energy density. The satiation ratio of a standard portion of almonds (158 kcal), with an average eating rate of 5 g per minute would be 158/5 = 32 kcal/min. The higher the ratio, the lower the satiation capacity of the food per kcal, the more needs to be ingested of the food item to feel satiated [13].

The knowledge that is gained can also be better exploited by food industries that may develop new (versions of existing) food products with a harder structure relative to their caloric content. This represents a challenge as texture and taste are major drivers of palatability and food acceptance [41,111]. Recently, Bolhuis and Forde provided directions for food technologists to make their foods more satiating by changing food structures [28]. In addition to the prevention of weight gain, this knowledge may be used to develop food products that stimulate food intake in malnourished populations. 

Besides direct food manipulations, the way of eating may potentially also be modified to slow the rate of food intake. For example, using a spoon instead of a straw or eating with chopsticks instead of a fork. These relatively simple adjustments have shown to reduce food intake [23]. Digital tools have also been developed to slow eating rate. Such tools provide immediate feed-back on the rate of eating, such as the smart fork or Mandometer [112,113].

Finally, we advocate that foods with a harder texture (relative to their energy content) should be part of dietary guidelines to maintain a healthy body weight. Currently the Dutch Nutrition Center (‘Voedingscentrum’), the Dutch equivalent to the US MyPlate, a federal program of the US Department of Agriculture (USDA), already mentions that smoothies and juices should not be counted as a daily portion of fruit given that you feel less satiated after a smoothie compared to eating whole pieces of fruit (hard texture) [114]. Furthermore, in the Dutch dietary guidelines it is stated that sugar containing beverages should be limited as these calories do not satiate [114]. Similarly the HHS recommends fruits, especially whole fruits [115]. An important step in the right direction, however clear recommendations about the consumption of foods with a “low energy intake rate” should be added, that is foods that are relatively low in eating rate given the energy that they provide. Such a recommendation has the potential to contribute to the solution to the prevention of overeating and the consequent development of overweight and obesity. 

## Figures and Tables

**Figure 1 nutrients-13-01391-f001:**
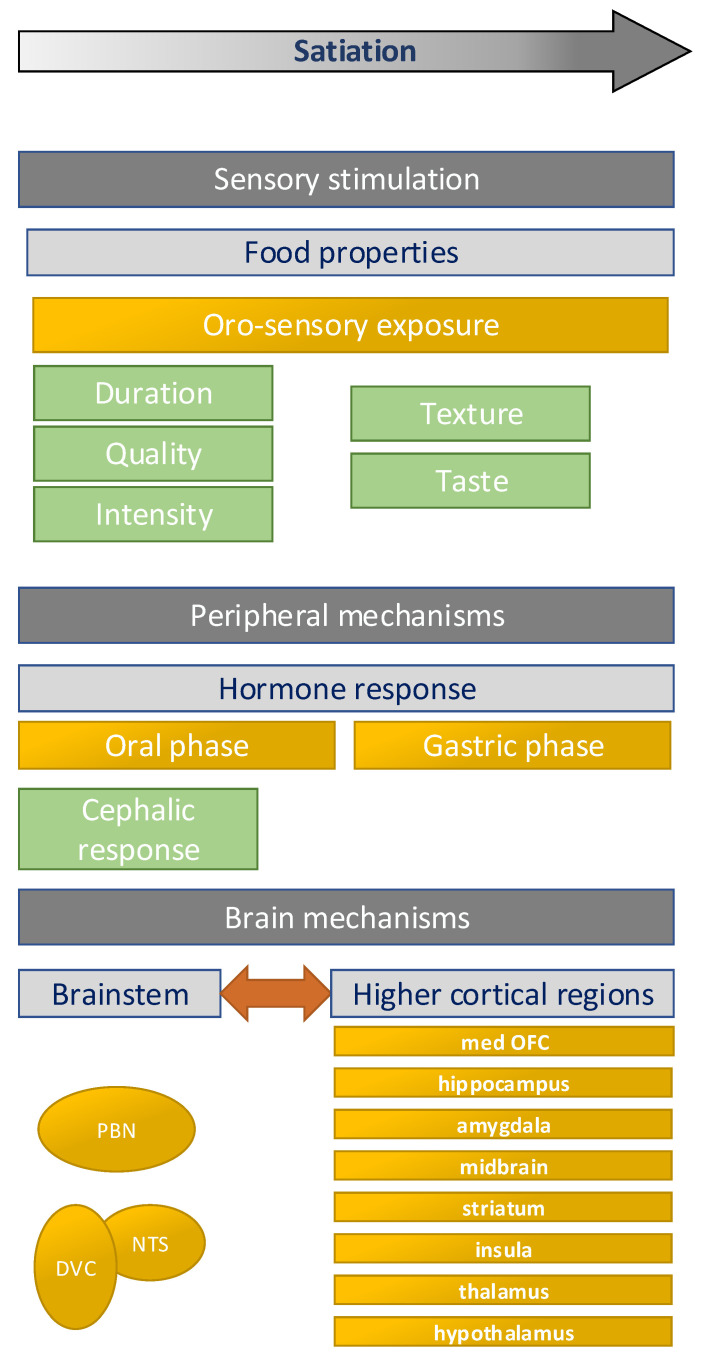
Overview of the factors involved in the relation between oro-sensory exposure and food intake. This figure is based partly on Hopkins and Blundell et al. [1,9]. PBN = parabrachial nuclei, NTS = nuclei of the solitary tract, DVC = dorsal vagal complex, med OFC = medial orbifrontal cortex.

**Figure 2 nutrients-13-01391-f002:**
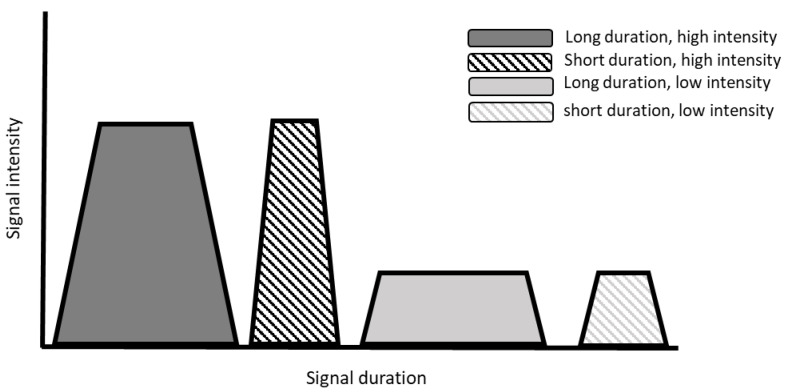
Schematic overview of our hypothesis underlying oro-sensory exposure. The area under the curve represents the level of oro-sensory exposure, which depend on both the intensity and the duration of the sensory signal.

**Figure 3 nutrients-13-01391-f003:**
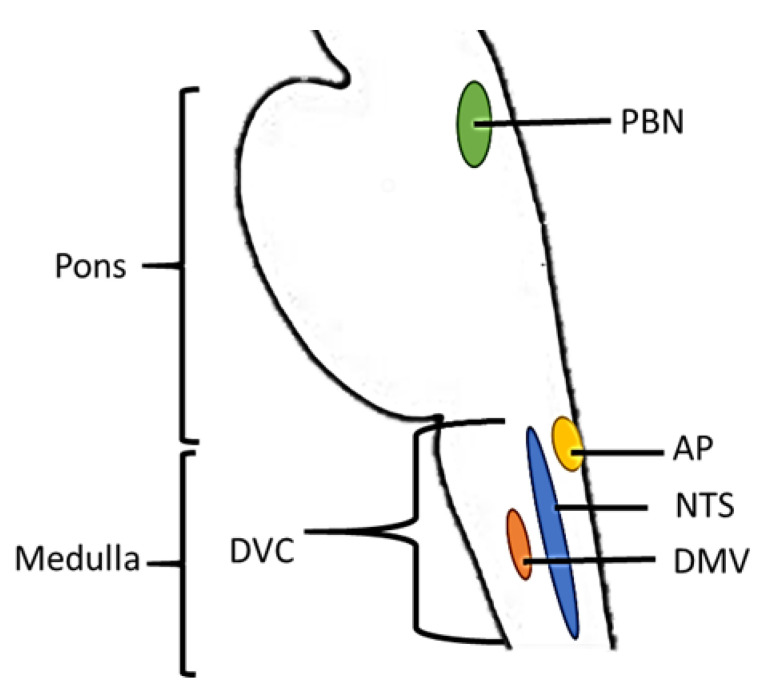
Graphical representation of a sagittal section of the brainstem. Depicted areas are involved in processing taste- and gastric signals and energy-related peripheral signals (hormones and metabolites). AP = Area postrema, DVC = dorsal vagal complex, DMV = Dorsal motor vagus, NTS = nucleus of the solitary tract, PBN = Parabrachial nuclei. Locations of brain areas are based on Duvernoy’s Atlas of the human brain stem [62].

**Figure 4 nutrients-13-01391-f004:**
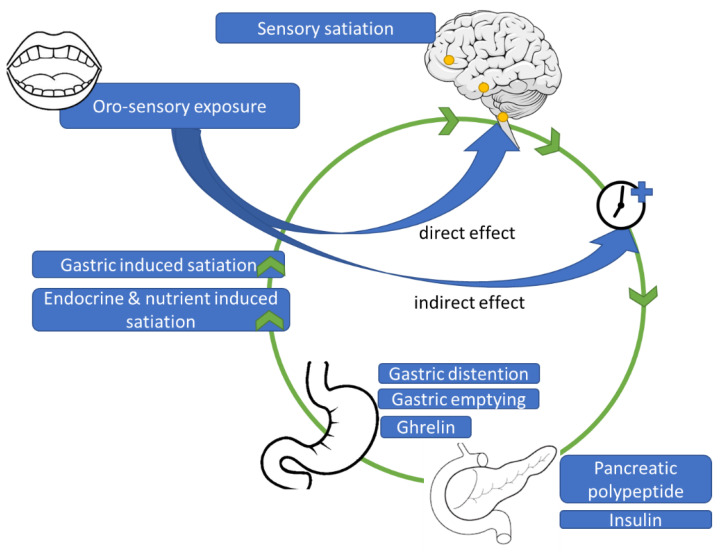
Proposed physiological mechanism underlying effects of oro-sensory exposure on satiation.

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
