# Peer review of "Effects of Oro-Sensory Exposure on Satiation and Underlying Neurophysiological Mechanisms—What Do We Know So Far?"

_nutrients, 2021, doi:10.3390/nu13051391_

Round 1
Reviewer 1 Report
The authors have prepared an interesting review of the effects of oro-sensory exposure on reaching satiety, or termination of eating within a meal. I do not have any major conceptual issues with the piece (though I did have a few specific questions, below). The paper is organized reasonably and flows well. The writing is understandable, but would benefit from a bit of editing (I have made some specific, though not exhaustive, suggestions below). There are some places where a bit more detail would be appreciated, including the relative effects of aroma and oro-sensory stimulation, but references are provided and I think adequate.
47-50: Sentence is a bit awkward, please consider rewording.
59-61: Interesting phrasing, would generally think of eating as required for homeostasis and not a threat, but there are potential threats too.
63-64: Incomplete sentence.
88: Medial instead of medical?
For section 2, it would be desirable to have some idea of the size of effects of oro-sensory exposure. How many calories does increased oro-sensory exposure save?
267-271: What about variety within a meal? Beyond the scope of the paper? Still, if the sensory specific satiety hypothesis is correct, one might expect oro-sensory exposure intensity to affect just the consumption of the food in question, and might even make people eat more of other items in the meal. What would that do for total calories consumed? Have the model meals in these studies all been very simple?
276: And cranial nerve V.
340 Diverge rather than dissect?
360: Ambiguous rather than unambiguous?
375-376: “Examples of hormones that are often names in this respect…” might read better as “Relevant hormones include…”
379: “decreased appetite responses and weight gain” This seems contradictory.
398: taught rather than learned?
399-401: Sentence needs some editing
475-479: “…co-manipulation effects such as changes in other sensory properties or energy density. This would prevent studies from describing the obtained results to specific manipulations while in fact the results may in part be explained by co-manipulation effects and are thus biased.” I think you might replace all this text with the word “confounds.”
Author Response
Reviewer 1
The authors have prepared an interesting review of the effects of oro-sensory exposure on reaching satiety, or termination of eating within a meal. I do not have any major conceptual issues with the piece (though I did have a few specific questions, below). The paper is organized reasonably and flows well. The writing is understandable, but would benefit from a bit of editing (I have made some specific, though not exhaustive, suggestions below). There are some places where a bit more detail would be appreciated, including the relative effects of aroma and oro-sensory stimulation, but references are provided and I think adequate.
Answer: Thank you for your feedback; we have revised the manuscript according to your suggestions.
47-50: Sentence is a bit awkward, please consider rewording.
Answer: thank you for pointing this out, we have rephrased the sentence into a few shorter sentences instead of 1 long sentence.
59-61: Interesting phrasing, would generally think of eating as required for homeostasis and not a threat, but there are potential threats too.
Answer: Thank you, it is a statement described by Woods 1991 where he describes the paradox of eating.
63-64: Incomplete sentence.
Answer: Rephrased, we now added the definition of sensory-specific satiety.
88: Medial instead of medical?
Answer: thank you for pointing that out, we changed it to medial.
For section 2, it would be desirable to have some idea of the size of effects of oro-sensory exposure. How many calories does increased oro-sensory exposure save?
Answer: depending on the intensity and the texture hardness it could be up to 30% reduction in food intake (grams). We added the following sentence (line 239) “ A review by Mc Crickerd et al. argues that texture effects on food intake are approximately (30 %) and taste effects (10%), depending on the difference in texture hardness and taste intensity between two eaten foods [13].”
267-271: What about variety within a meal? Beyond the scope of the paper? Still, if the sensory-specific satiety hypothesis is correct, one might expect oro-sensory exposure intensity to affect just the consumption of the food in question, and might even make people eat more of other items in the meal. What would that do for total calories consumed? Have the model meals in these studies all been very simple?
Answer: this is indeed beyond the scope of the paper, we chose to include only studies with single “simple” foods as the scope of this review is oro-sensory exposure and satiation, and should fit within a special issue that focusses on satiety and appetite control- gut mechanisms. Variety within a meal is a topic on its own.
276: And cranial nerve V.
Answer: that is indeed a mistake, thank you for pointing this out, we have added it.
340 Diverge rather than dissect?
Answer: we changed it to diverge
360: Ambiguous rather than unambiguous?
Answer: thank you, we changed to ambiguous
375-376: “Examples of hormones that are often names in this respect…” might read better as “Relevant hormones include…”
Answer: “changed to “ examples of cephalic hormones are...” line 380.
379: “decreased appetite responses and weight gain” This seems contradictory.
Answer: we see your point, changed it to “ appetite regulation” line 385
398: taught rather than learned?
Answer: we changed to "taught".
399-401: Sentence needs some editing
Answer: We changed the sentence to: “Especially neuroimaging studies have taught us more about the processing of sensory signals in the higher cortical regions of the brain in relation to satiation. However, the exact neurophysiological mechanism of how oro-sensory signals affect the process of satiation, and what the role of the brain stem is in this still needs to be elucidated.”
475-479: “…co-manipulation effects such as changes in other sensory properties or energy density. This would prevent studies from describing the obtained results to specific manipulations while in fact the results may in part be explained by co-manipulation effects and are thus biased.” I think you might replace all this text with the word “confounds.”
Answer: Thank you for the suggestion, we changed the text to: “ Therefore future research studies should better define the manipulation of interest while minimizing confounding due to changes in other sensory properties or energy density.” Line 479
Reviewer 2 Report
This is a very useful, well-organized and generally well-written review of an important topic. I recommend publication subject to some very minor changes, mostly to do with English expressions; as follows.
Lines 32-33. This claim ignores the important role of contextual factors e.g. portion size, watching TV, in how much of a meal people eat. The authors need to point out that the research they review holds such factors constant.
Line 152: should be ‘processed’
Line 178: surely ‘larger’ should be replaced by ‘smaller’?
Line 222: ‘As mentioned before, ..’ not: ‘Like mentioned before ..
Line 285: ‘many’ instead of ‘quite some’
Lines 399-402: The sentence starting with ‘However’ needs to be re-written
Line 439: insert ‘hormonal responses’ instead of ‘responses in hormones’
Line 507: instead of ‘the development overweight’ perhaps ‘yet further increases in the bodyweights of future generations’
Author Response
Reviewer 2
This is a very useful, well-organized and generally well-written review of an important topic. I recommend publication subject to some very minor changes, mostly to do with English expressions; as follows.
Answer: thank you for your nice words
Lines 32-33. This claim ignores the important role of contextual factors e.g. portion size, watching TV, in how much of a meal people eat. The authors need to point out that the research they review holds such factors constant.
Answer: As this review is written for a special issue that focuses on gut mechanisms affecting appetite control, specifying the contextual factors would be out of the scope of the issue.
Line 152: should be ‘processed’
Answer line 158: Adjusted to processed
Line 178: surely ‘larger’ should be replaced by ‘smaller’?
Answer: Thank you for pointing out this mistake, we changed it accordingly.
Line 222: ‘As mentioned before, ..’ not: ‘Like mentioned before ..
Answer: we revised the sentence.
Line 285: ‘many’ instead of ‘quite some’
Answer: we changed the wording as suggested
Lines 399-402: The sentence starting with ‘However’ needs to be re-written
Answer: we have re-written the sentence into: “ However, the exact neurophysiological mechanism of how oro-sensory signals affect the process of satiation, and what the role of the brain stem is in this, still need to be elucidated “.
Line 439: insert ‘hormonal responses’ instead of ‘responses in hormones’
Answer: we have changed it to "hormonal responses"
Line 507: instead of ‘the development overweight’ perhaps ‘yet further increases in the bodyweights of future generations’
Answer: thank you for this suggestion, the sentence has been changed accordingly
Reviewer 3 Report
In general, there are a couple of areas where the manuscript has been very well written and some areas that are rather confusing. There is a mention in children in further work and a couple of other suggestions for further work but these were not part of the main manuscript which makes it difficult to understand why they would be useful areas of study when either the mechanisms or relation to oro-sensory. Additionally, the manuscript tense needs to be revised accordingly as there are some places where the tenses are used wrongly and this does affect the flow of the paper.
A couple of areas to consider for revision are listed below:-
L110, Please separate isimportant into two words.
Figure 2, what is the source of the information that was used to plot that graph?
L222: Please revise as like mentioned before does not read well.
L224: Could the authors please expound on what they mean by the following statement? “by prolonging the exposure” this is rather incomplete statement.
L236 What do the authors mean by the statement “…how much people consume from it”
L247: please rewrite the sentence to make it clearer
L249 What does “…saltiness 248 and sweetness this is possible” mean?
L261 predictor should be predictors
Figure 3 and 4: Did the authors draw the image by themselves or not? If no, do the authors have permission to reuse the image, or is it freely available to reproduce on creative commons? If yes, could they include the license, please?
The authors need to decide whether they want to use sated or satiated and not using this interchangeably.
L376 Please change names to named
L379 –onwards please revise the tense accordingly as I would imagine that you are currently hypothesizing and not in the past?
L393 Do you mean rather than or other than? It is not clear based on the context of what is being discussed.
L398-401 Please revise
L562 something is missing in the sentence
Author Response
Reviewer 3
In general, there are a couple of areas where the manuscript has been very well written and some areas that are rather confusing. There is a mention in children in further work and a couple of other suggestions for further work but these were not part of the main manuscript which makes it difficult to understand why they would be useful areas of study when either the mechanisms or relation to oro-sensory. Additionally, the manuscript tense needs to be revised accordingly as there are some places where the tenses are used wrongly and this does affect the flow of the paper.
Answer: Thank you for your compliments and your thorough review.
A couple of areas to consider for revision are listed below:-
L110, Please separate isimportant into two words.
Answer: thank you for pointing out this error, we have separated the two words.
Figure 2, what is the source of the information that was used to plot that graph?
Answer: this conceptual figure is used to illustrate the concept of oro-sensory exposure and is our own hypothesis based on the literature reviewed in this review. To underline that it is our hypothesis we changed the caption of the figure to “Schematic overview of our hypothesis of the concept of oro-sensory exposure. The area under the curve represents the level of oro-sensory exposure, it depends on both the intensity and the duration of the sensory signal”
L222: Please revise as like mentioned before does not read well.
Answer: we adjusted the sentence: “ The concentration of a tastant in a food product does not predict intensity per se as the release of tastes and flavors also depends on the structure of the food” line 224
L224: Could the authors please expound on what they mean by the following statement? “by prolonging the exposure” this is rather incomplete statement.
Answer: we agree that this is not clear, we rephrased it to: “ It is shown that a harder structure enhances the effect of intensity, by prolonging the exposure duration (i.e. harder textures need longer oral processing before a bolus is safe to swallow)”
L236 What do the authors mean by the statement “…how much people consume from it”
Answer: changed to: “ The taste intensity of foods and how they are being processed in the mouth affects how much food people consume”
L247: please rewrite the sentence to make it clearer
Answer: we have revised the order of the sentences and the paragraph reads more clear now.
L249 What does “…saltiness and sweetness this is possible” mean?
Answer: It refers to the beforementioned sentence, combined the two sentences to make it more clear. Line 253: “ Critical in studies investigating oro-sensory exposure and food intake is that the palatability of the foods should be equal. You can study two suboptimal perceived intensities that are equally liked, such as shown in the studies described before on sweet taste and salt taste intensity [12, 37].”
L261 predictor should be predictors
Answer: thank you for pointing that out, we have changed it
Figures 3 and 4: Did the authors draw the image by themselves or not? If no, do the authors have permission to reuse the image, or is it freely available to reproduce on creative commons? If yes, could they include the license, please?
Answer: We drew both Figures 3 and 4 ourselves, so we do not need permission to reuse the image.
The authors need to decide whether they want to use sated or satiated and not using this interchangeably.
Answer: we changed all to satiated
L376 Please change names to named
Answer: this has been adjusted
L379 –onwards please revise the tense accordingly as I would imagine that you are currently hypothesizing and not in the past?
Answer: Based on our recent systematic review we do no longer hypothesize that cephalic phase hormone responses have a clear function therefore we made this paragraph past tense. We now added “ up to recently” to clarify this and changed concluded to we conclude.
L393 Do you mean rather than or other than? It is not clear based on the context of what is being discussed.
Answer: Changed to “indistinguishable from normal variation”.
L398-401 Please revise
Answer: revised to: “To summarize, the neurophysiological mechanisms underlying satiation are gradually becoming clearer as research progresses. Especially neuroimaging studies have taught us more about the processing of sensory signals in the higher cortical regions of the brain in relation to satiation.” Line 402
L562 something is missing in the sentence
Answer: we rephrased it to the following ”Additionally, in the Dutch dietary guidelines it is stated to limit sugar-containing beverages as these calories do not satiate [119].” Line 564